# Evidence Supporting the Regulatory Relationships through a Paracrine Pathway between the Sternum and Pectoral Muscles in Ducks

**DOI:** 10.3390/genes12040463

**Published:** 2021-03-24

**Authors:** Yanying Li, Hehe Liu, Lei Wang, Yang Xi, Jiwen Wang, Rongping Zhang, Liang Li, Lili Bai, Ahsan Mustafa

**Affiliations:** 1Farm Animal Genetic Resources Exploration and Innovation Key Laboratory of Sichuan Province, College of Animal Science and Technology, Sichuan Agricultural University, Chengdu 611130, China; sicauliyanying@163.com (Y.L.); hhd199312138@163.com (L.W.); xiyang9596@163.com (Y.X.); wjw2886166@163.com (J.W.); dkzhrp@126.com (R.Z.); LL457@163.com (L.L.); bll20ran@163.com (L.B.); 2Institute of Animal Nutrition, Key Laboratory for Animal Disease-Resistance Nutrition of China, Ministry of Education, Sichuan Agricultural University, Chengdu 611130, China; dr.ahsan.mustafa@gmail.com

**Keywords:** duck, RNA-seq, sternum, pectoral muscle, interaction

## Abstract

Muscles and bones are anatomically closely linked, and they can conduct communication by mechanical and chemical signals. However, the specific regulatory mechanism between the pectoral muscle and sternum in birds was largely unknown. The present study explored the potential relationship between them in ducks. The result of the sections showed that more nuclei in proliferate states were observed in the pectoral muscle fibers attached to the calcified sternum, than those attached to the un-calcified sternum. The RNA-seq identified 328 differentially expressed genes (DEGs) in the sternum between the calcified and un-calcified groups. Gene ontology (GO) showed that the DEGs were mainly enriched in pathways associated with calcification. In addition, DEGs in the muscles between the calcified and un-calcified sternum groups were mainly annotated to signal transduction receptor pathways. The expression patterns of genes encoding for secreted proteins, in bone (CXCL12, BMP7 and CTSK) and muscle (LGI1), were clustered with muscle development (MB) and bone calcification (KCNA1, OSTN, COL9A3, and DCN) related genes, respectively, indicating the regulatory relationships through a paracrine pathway existing between the sternum and pectoral muscles in ducks. Together, we demonstrated that the pectoral muscle development was affected by the sternal ossification states in ducks. The VEGFA, CXCL12, SPP1, NOG, and BMP7 were possibly the key genes to participate in the ossification of the duck sternum. We firstly listed evidence supporting the regulatory relationships through a paracrine pathway between the sternum and pectoral muscles in ducks, which provided scientific data for the study of the synergistic development of bone and skeletal muscle.

## 1. Introduction

The bone and muscle are two independent secretion organs and the largest secretion organs (endocrine function), which are synergistic in vertebrate development [1]. The bone provides an attachment, and serves as a lever for skeletal muscle movement, whereas the contraction of the skeletal muscle also affects the skeletal structure and contributes to the increase in bone strength [2]. Medical studies have found that, in the case of aging or disease, bone and muscle tend to degenerate simultaneously. Most osteoporosis patients are accompanied by sarcopenia, while the patients with long-term bedridden muscle atrophy easily develop osteopenia [3]. Hence, investigating the coordination development process between bone and skeletal muscle would be a new perspective on the therapy of human sarcopenia and osteopenia.

Bones and muscles can synthesize and secrete a variety of cytokines, which can participate in the perception and transmission of signals, such as mechanical stretch stimulation and changes in nutrient levels, as well as a mutual influence on each other’s development process and states [3]. For example, myostatin, which is a muscle-secreted protein, can regulate bone remodeling by stimulating osteoclast differentiation directly [4], whereas osteocalcin, secreted from osteoblasts, can increase nutrient uptake and catabolism of muscle fibers during exercise [5]. Thus, the paracrine pathway may play a central effect in mediating the coordination development process between bone and skeletal muscle.

In poultry production, the pectoral muscle, which is attached to the sternum, is considered an important economic trait [6]. Our previous studies have shown that there was a synergistic development between the sternum and pectoral muscles in ducks. Firstly, the area and height of the sternum determine the width and thickness of the pectoral muscle, thus affecting the output of the pectoral muscle. Secondly, the pectoral muscle remained in a state of development before sternal ossification. At the age of 8 to 9 weeks, the duck sternum was completely ossified, and then the pectoral muscle gradually stopped further developing [7]. It was implied that sternal development states may affect pectoral muscle development in ducks.

Although some progress has been made in the regulatory relationships between bone and muscle, especially in human sarcopenia and osteopenia, it was still unclear in birds. The morphology of the sternum in birds is different from that in mammals, as the ossification of bird sternums does not initiate in multiple separate regions as observed in the mammal [7]. In flying birds, the sternum is one of the most important and characteristic bone elements, which is highly adapted to flight, so its keel is extremely prominent [8]. The sternum of flying birds provides attachment surfaces for strong flight muscles, including the *m. supracoracoideus* and *m. pectoralis*; the former is particularly important during take-off, while the latter is mainly responsible for the downstroke [9]. In poultry, there is no abnormal protruding keel on the sternum, and the sternum tends to be flat. The area of the sternum directly determines the output of the pectoral muscle. Therefore, the present study combined tissue sections with transcriptome data to explore the potential connections between bone and muscle. This study would be helpful in listing more insights about the interaction between bone and muscle and laying a foundation for the molecular-assisted breeding of ducks to enhance the breast meat yield and early breast meat development rate.

## 2. Materials and Methods

### 2.1. Animals and Sample Collection

The ducks (NongHua-GF2 strain) used, in the present study, were provided by the Waterfowl Breeding Experimental Farm of Sichuan Agricultural University in Ya’an, China. For the RNA-seq, three male ducks with similar body weights at the age of 6 weeks old were selected for sampling. The calcified and un-calcified sternums, as well as the pectoral muscles attached to them, were taken as samples, respectively (Figure 1). For hematoxylin-eosin (H&E) staining, three male ducks were selected as samples at 6 and 7 weeks, respectively. The junction of the sternum (calcified and un-calcified) and the pectoral muscle was sampled as a solid whole, and the size of all samples was controlled around 1 × 0.5 cm. Then, the samples were fixed in a 10% formalin solution (Solarbio, China) for at least 24 h. Moreover, the remaining sternums and muscles were ground with liquid nitrogen and stored at −80 °C. All experimental procedures were carried out in strict accordance with the regulations of the animal ethical and welfare committee (AEWC) of Sichuan Agricultural University China (approval code: AEWC2016, 6 January 2016).

### 2.2. Hematoxylin and Eosin Staining Assay

The fixed samples were dehydrated through an ethanol series, embedded in paraplast and cut into 5 μm sections as described. Then the sections were stained with H&E (Servicebio, Wuhan, China), dehydrated, and then covered with neutral balsam. The rest sections were collected for immunofluorescence staining assays.

### 2.3. Immunofluorescence Staining Assays

The proliferating cell nuclear antigen (PCNA) was used to test proliferating cells. The sections were dewaxed, rehydrated, and then incubated with EDTA (pH 9.0; Servicebio, China) for 20 min at 100 °C. After washing with phosphate-buffered saline (PBS) three times, sections were incubated with 3% Bovine Serum Albumin (BSA) (Solarbio, China) for 30 min at room temperature. Then, sections were incubated with a primary mouse PCNA antibody (1: 100; Servicebio, China) at 4 °C overnight, and incubated with goat anti-mouse IgG labeled with red Cy3 (1: 300; Servicebio, China) for 50 min at room temperature. Finally, sections were counterstained with DAPI (Servicebio, China). Photographs were digitized by using a fluorescent microscope equipped with an imaging system (Nikon, Tokyo, Japan).

### 2.4. RNA Extraction, Library Construction, and Sequencing

RNA was extracted using RNAiso Plus (Takara, Shiga, Japan). The RNA concentration of each sample was measured using a NanoDrop 2000 Spectrophotometer (Thermo Fisher Scientific, Wilmington, DE, USA), while RNA integrity was measured using an Agilent 2100 Bioanalyzer (Agilent Technologies, CA, USA). High-quality RNA was sent to Biomarker Technologies Corporation (Beijing, China) for cDNA libraries construction and sequencing. The cDNA libraries were prepared with multiplexing primers using the NEB Next^®^ Ultra RNA Library Prep Kit for Illumina (New England Biolabs, Ipswich, MA, USA). The libraries were constructed with average inserts of 200 bp (150~250 bp), with non-stranded library preparation (Figure 1). The raw transcriptome read data are available in the SRA database with accession number PRJNA605777 (https://www.ncbi.nlm.nih.gov/bioproject/PRJNA605777/, accessed on 5 March 2021).

### 2.5. Functional Annotation and Differentially Expressed Genes (DEGs)

Firstly, genome files (duckbase.refseq.v4.fq) and genome annotation files (duckbase.refseq.v4.NCBI2016.gtf) were downloaded from NCBI. Hisat2 (v2.1.0) built the index of genome files and mapped RNA-seq data to the reference genome. Next, aligned BAM files were indexed and sorted with SAMtools (V0.1.19-44428cd). StringTie (v1.3.3b) assembled the alignments into full and partial transcripts, estimated the expression levels of all genes and transcripts, and created read in files for Ballgown. Finally, the software packages in R (v3.5.1), such as gene filter, dplyr, and devtools, were used to analyze the differential genes.

### 2.6. Gene Ontology (GO) and Kyoto Encyclopedia of Genes and Genomes (KEGG) Enrichment Analysis

Functional groups and pathways encompassing the DEGs were identified based on the GO and KEGG pathway analysis using the Database for Annotation, Visualization, and Integrated Discovery (DAVID v.6.8) software. The threshold was set as a modified Fisher Exact *p*-value ≤ 0.05.

### 2.7. Protein Interaction Network Analysis

The analysis of protein interaction networks was performed using the STRING protein database (http://string-db.org/, accessed on 11 October 2019). Visual editing of the differential gene-protein interaction network data files was performed by Cytoscapes of ware version 3.6.1 (http://www.cytoscape.org/, accessed on 16 October 2019). Then, the plugin Molecular Complex Detection (MCODE) was applied to screen the modules of the protein–protein interaction (PPI) network in Cytoscape.

### 2.8. Real-Time PCR

Real-time PCR was performed to validate RNA-seq results. The total RNA from 12 samples was extracted using RNAiso Plus. The cDNA was synthesized using the reverse transcript system (Takara, Japan) according to the manufacturer’s instructions. Real-time PCR was carried out with the SYBR PrimeScript RT-PCR Kit (TaKaRa, Japan) using the Bio-Rad CFX Manager (Bio-Rad Laboratories, Berkeley, CA, USA). GAPDH and β-actin were used as the housekeeping genes for normalization of the gene expressions in all samples. The primers used for the interested and housekeeping genes in this study are listed in Appendix A.

### 2.9. Statistical Analysis

The 2^-△△^CT method [10] was employed to calculate the relative mRNA expression of interested genes according to the Ct value, and the results were normalized by using the internal control genes β-actin and GAPDH. The entire analysis was performed with the SAS software package, version 8.2 (SAS Institute, Cary, NC, USA, 1999). Results are formatted as means ± standard error (S.E.). A *p*-value of ≤0.05 was considered statistically significant.

## 3. Results

### 3.1. Histological Observation of Sternum and Pectoral Muscle from Ducks

When the ducks were at six weeks, the H&E staining showed that the un-calcified sternum was full of non-hypertrophic chondrocytes, which were small, flat, and sparsely arranged. In the calcified sternum, some of the cells were large and round, indicating that hypertrophic chondrocytes had appeared. When the ducks were at seven weeks, the vascular and osteoblasts appeared in the sternum, indicating that ossification had occurred (Figure 2). Additionally, more nuclei were observed in the muscle growth on the calcified sternum than that attached to the un-calcified sternum. The immunofluorescence staining assay was conducted in the pectoral muscle using anti-PCNA, and results showed that the positive signal of PCNA in the calcified group was more than that in the un-calcified group of pectoral muscles in ducks (Figure 3).

### 3.2. Transcriptome Analysis and Functional Genes’ Annotation

The sequencing and assembly statistics are summarized in Appendix A. An average of 26 million reads in each sample were mapped to the duck reference genome (ID: PRJNA605777). The correlation coefficient of the gene expression levels, based on the FPKM values, among samples, was higher than 0.89 (Figure 4A), indicating that the selection of experimental samples was consistent and reliable. Next, we screened the DEGs in the sternum between the calcified and un-calcified groups, according to the threshold set to “*p* < 0.05, |Log2FC| > 1”. A total of 328 genes, with 238 up-regulated and 90 down-regulated, were identified as DEGs (Appendix A). In pectoral muscles between the calcified and un-calcified groups, a total of 12 significantly down-regulated genes and 23 significantly up-regulated genes were identified (Figure 4B).

### 3.3. Functional Enrichment of DEGs

We performed a GO analysis of DEGs identified in the sternum (Figure 5A). Within the biological process category, genes involved in chemotaxis, branching involved in salivary gland morphogenesis, and integrin-mediated signaling pathways were counted as the top three categories. Furthermore, biological processes of enrichment include cell-matrix attachment positive regulation of bone resorption, BMP signaling pathways, embryonic limb morphogenesis, and negative regulation of chondrocyte differentiation. In the cellular component category, genes belong to extracellular space, an integral component of the membrane, and the extracellular region was highly enriched. In the molecular function category, GTPase activator activity, calcium ion binding, and transporter activity accounted for a major proportion. The KEGG enrichment showed that 12 significantly enriched signaling pathways were enriched, based on the DEGs in the sternum (*p* < 0.05). These pathways included ECM-receptor interaction, cytokine-cytokine receptor interaction, Focal adhesion, etc. (Figure 5B).

In pectoral muscles between the calcified and un-calcified groups, GO terms relative to non-membrane spanning protein tyrosine kinase activity, calcium ion binding, and cardiac muscle tissue morphogenesis were counted as the top enriched categories. The KEGG enriched five significantly signaling pathways (*p* < 0.05), and among them, the calcium signaling pathway enriched the greatest number of DEGs (Figure 5C).

### 3.4. The Potential Relationships between the Sternum and Pectoral Muscles

The bone is an endocrine organ, as it can affect the calcification process through secretory proteins [11]. We enriched the genes encoding for secretory proteins based on the DEGs by the GeneCards database (https://www.genecards.org/, accessed on 14 October 2019). A total of 195 genes were screened in the sternum between the calcified and un-calcified groups. Among them, some are involved in the osteocrine process, such as DKK1, VEGFA, SOST, BMP3/6/7, GDF5, BGLAP, OSTN, IHH, OGN, NOG, SPP1, COL10A1, PTHLH, etc. Six of them were selected for gene expression determination by real-time PCR, and the results revealed their expression trends exhibited significantly similarly (r^2^ = 0.7147) with the RNA-Seq data (Appendix A), indicating that the RNA-Seq data were reliable (Figure 6).

Next, we used MCODE to screen the modules of the PPI network using genes encoding for secretory proteins (Appendix A). The genes involved in the top two significant modules, which may play a more important role in the PPI network, were therefore used for the subsequent enrichment analysis. The results showed that the DEGs in modules 1 and 2 were principally related to focal adhesion, ECM-receptor interaction, TGF-β signaling pathway, and cytokine-cytokine receptor interaction (Appendix A).

In order to further explore the potential roles of the paracrine pathway in mediating the internal relationships between bone and muscle, we performed a PPI analysis based on the genes encoding for the secreted proteins screened in the sternum and all DEGs in the pectoral muscle (|Log2FC| > 1). According to the data obtained on the STRING (Appendix A), the top three high degree hub nodes, including IGF1, VEGFA, and CXCL12, were identified and visualized by the Cytoscape platform software. IGF1 can directly interact with eight DEGs in the pectoral muscle, including MB, MRC1, DNTT, FBXO32, and PTK2. VEGFA can directly interact with seven DEGs in the pectoral muscle, including PTK2B, WASF1, EGLN3, DNTT, and MRC1. CXCL12 can directly interact with GNAT3, LCK, MRC1, GRM4, EGF, and MB in the pectoral muscle (Figure 7A).

Similarly, a total of 15 genes, which encode for secretory proteins, were found in the DEGs pectoral muscles between the calcified and un-calcified groups by the GeneCards database (https://www.genecards.org/, accessed on 14 October 2019). The PPI analysis was performed based on these 15 genes, and all DEGs in the bone were analyzed (Appendix A). The results showed that COL6A6, PCSK2, and LGI1 may be in the central regulatory position of the interaction network. COL6A6, secreted by the bone, can directly interact with DEGs in the pectoral muscle, such as ITGB3, COL10A1, COL9A3, and COL4A3/4. Besides, the secretory protein LGI1 was supported to interact with KCNA1, GABRG2, CNTN2, and GABRA1 by STRING (Figure 7C).

Their FPKMs, in tissues, were then performed for the gene expression cluster analysis. It was found that protein-coding genes in the sternum, such as DKK1, BMP7, CTSK, and CXCL12, can be clustered with MB genes in the muscle (Figure 7B). In addition, the protein-coding gene LGI1 in the muscle was clustered with the bone differential gene KCNA1 (Figure 7D). The expression clustering information between other DEGs and secretory proteins are arranged in Appendix A. Then, we used RT-PCR to further detect the expression patterns of these genes. The results showed that protein-coding genes in the sternum (CTSK and CXCL12) and DEGs in the muscle (MB) had the same expression patterns. Moreover, there is the same expression pattern between secretory proteins in the muscle (LGI1) and DEGs in the bone (KCNA1). These results were consistent with those obtained by the transcriptome (Figure 8).

## 4. Discussion

The ossification pattern of the bird sternum belongs to endochondral ossification [8]. The endochondral ossification was comprised by a series of steps, including the proliferation and differentiation of chondrocytes, angiogenesis, vascular invasion, and mineralization of the bone matrix [8,12,13]. In the process of ossification, the transformation of the cell type, tissue structure, physiological and biochemical characteristics were completed, and finally hard bone was formed [14]. In this study, H&E staining showed that chondrocytes gradually become hypertrophic during the process of sternal calcification in ducks. It was reported that hypertrophic chondrocytes can secrete vascular endothelial growth factor (VEGF) and nuclear factor-kB receptor activating factor ligand (RANKL) [15], to promote the distribution of vascularity in the region of hypertrophic chondrocytes [16]. From the section results, with the gradual development of the sternum, a small number of blood vessels began to appear, indicating that hypertrophic chondrocytes caused vascular invasion. Since then, it was observed that vascular invasion brings osteoblasts, and blood cells continue to increase. The basic process of sternal ossification in ducks was consistent with the occurrence of endochondral ossification.

Based on this, we took the sternum and its corresponding muscles for RNA-seq. The KEGG enrichment based on the DEGs in the bones between the calcified and un-calcified groups demonstrated that the BMP signaling pathway, embryonic limb morphogenesis, negative regulation of chondrocyte differentiation, and negative regulation of angiogenesis were related to sternal development. Among the DEGs, SHH [17], IHH [18], PTHLH [19], GDF5 [20], SOX9 [21], BMP7 [22] and NOG [23] were reported to regulate the proliferation and differentiation of chondrocytes. Besides, LECT1 [24], TNMD [25], THBS1 [24] and VASH1 [26] contributed to angiogenesis and vascular invasion before the replacement of cartilage by bone during endochondral bone development, which was consistent with our findings of vascular appearance by H&E staining.

Moreover, the pectoral muscle development in the duck was induced during the calcification process of the sternum, reflected by the number of nuclei with the positive signals of PCNA. The sternum, as an endocrine organ, could affect muscle development by secretory proteins through the paracrine pathway during the process of ossification [1]. In this study, we found that 71 differential genes, accounting for 40% of the total number of genes, were enriched in the top three pathways (ECM-receptor interaction, cytokine-cytokine receptor interaction, and focal adhesion). These pathways were reported to be associated with the growth, proliferation, differentiation, and apoptosis of a variety of cells. Among them, cytokines bind to their specific receptors expressed on the membrane surface of cells, thereby leading to receptor oligomerization and the activation of intracellular signaling cascades [27]. It was implied that cytokines secreted by bones may affect muscle development by binding to receptors on the surface of muscle cells.

CXCL12, a protein secreted by bones, plays a role in various tumorigenic processes through the paracrine pathways, including cell growth [28,29], metastasis [30], and angiogenesis [31]. Vascular endothelial growth factor-A (VEGFA), a major angiogenic growth factor, upregulates MB in ischemic muscles, both in-vitro and in-vivo. MB is expressed in skeletal and cardiac muscles, which facilitates the movement of oxygen within muscles [32]. Studies have shown that the positive impact of CXCL12 on muscle regeneration is related to satellite cells and myoblasts [33]. Besides, our study found that CXCL12 was linked with the muscle differential gene MB, implying the regulatory roles of CXCL12 and MB in the sternum and pectoral muscles of ducks. In addition, LGI1, a protein secreted by muscles, may regulate the activity of voltage-gated potassium channels and be involved in neuronal growth regulation and cell survival [34]. KCNA1 encodes a voltage-gated delayed potassium channel. Mutations in the KCNA1 causes episodic ataxia type 1, which is characterized by interepisodic muscle weakness [35]. Recently, direct evidence identified the motor nerve as an important generator of myokymic activity and showed that the KCNA1 channel alters Ca2+ homeostasis in motor axons [36]. LGI1 can alter KCNA1 inactivation [37]. Our study also found that the mRNA expression trend of LGI1 was consistent with that of KCNA1, implying the regulatory roles of LGI1 and KCNA1 channels in the sternum and pectoral muscles of ducks. These findings support the regulatory relationships through the paracrine pathways that exist between the sternum and pectoral muscles in ducks.

## 5. Conclusions

In conclusion, we found that the pectoral muscle development was affected by sternal ossification states. RNA-seq results revealed that genes, such as VEGFA, CXCL12, SPP1, NOG, BMP7, etc., were screened as the key factors to participate in the ossification of the duck sternum. Moreover, secretory proteins CXCL12 and LGI1 may regulate the internal relationships between the bone and muscle through a new paracrine pathway. These findings would be helpful to understand the regulatory relationships between the bone and muscle and may provide a theoretical basis for improving poultry production.

## Figures and Tables

**Figure 1 genes-12-00463-f001:**
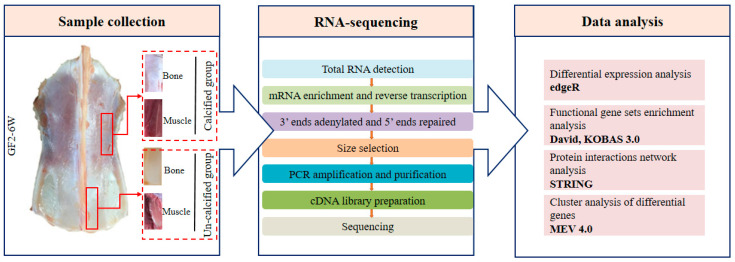
Sample collection and transcriptome data analysis process.

**Figure 2 genes-12-00463-f002:**
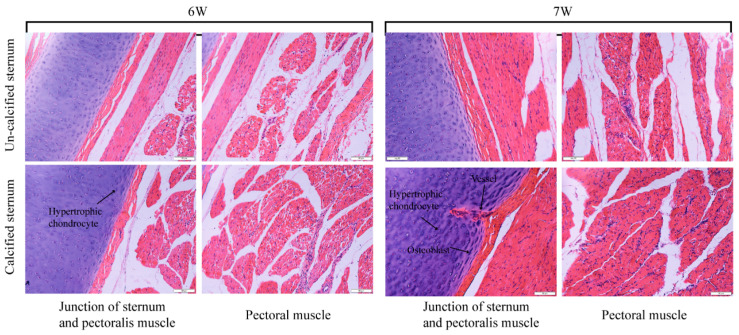
Histological observation of the development of the sternum and pectoral muscles in ducks. The samples were taken from the calcified and un-calcified sternum junction of the sternum and the pectoral muscle (Figure 1).

**Figure 3 genes-12-00463-f003:**
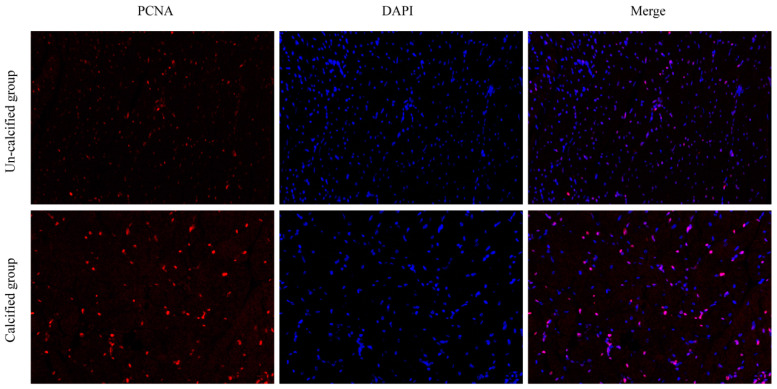
Immunofluorescence staining for Proliferating Cell Nuclear Antigen (PCNA) expression in the development of duck pectoral muscle.

**Figure 4 genes-12-00463-f004:**
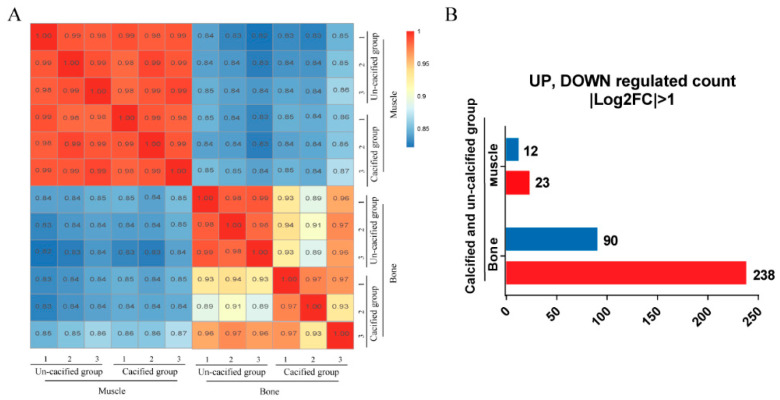
The correlation analysis among samples and the number of differentially expressed genes. (**A**) The correlation analysis among samples based on the differentially expressed genes. The horizontal and vertical coordinates are log10 (FPKM + 1) of the samples compared with each other. (**B**) The differentially expressed genes in the sternum and muscle between the calcified and un-calcified groups are shown as up-regulated or down-regulated, in which red represents up-regulated and blue represents down-regulated.

**Figure 5 genes-12-00463-f005:**
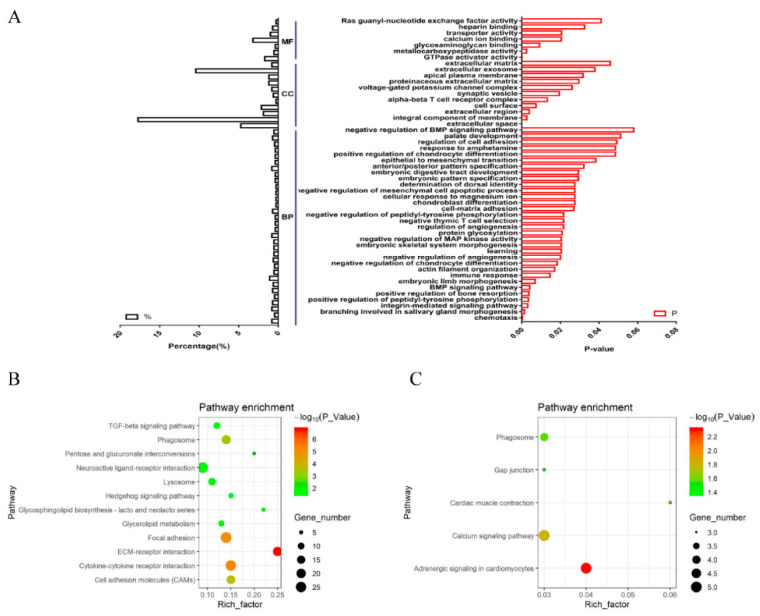
Enrichment analysis of DEGs between the calcified and un-calcified groups in the sternum and pectoral muscles. (**A**) Gene ontology (GO) annotation of differentially expressed genes (DEGs) in sternum. (**B**) Kyoto Encyclopedia of Genes and Genomes (KEGG) enrichment scatter plot of DEGs in the sternum. (**C**): KEGG enrichment scatter plot of DEGs in pectoral muscles. All DEGs in the sternum and pectoral muscles were compared between the calcified and un-calcified groups.

**Figure 6 genes-12-00463-f006:**
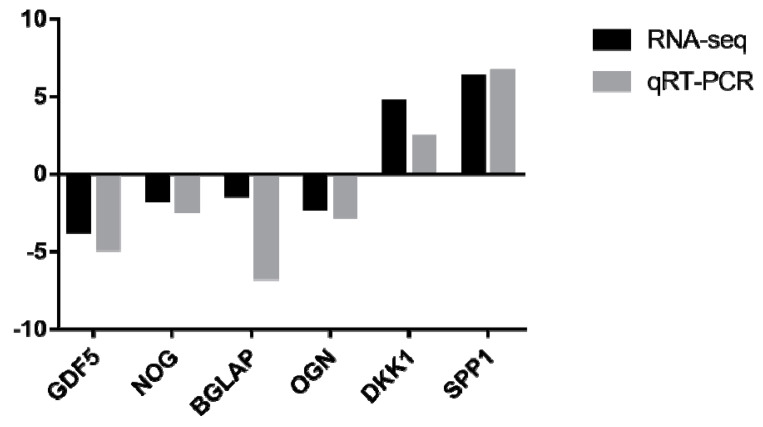
Comparison of mRNA expressions between real-time PCR and RNA-seq data. The gene expression values and Fragments Per Kilobase of exon model per Million mapped fragments (FPKM) values were transformed to log2 scale.

**Figure 7 genes-12-00463-f007:**
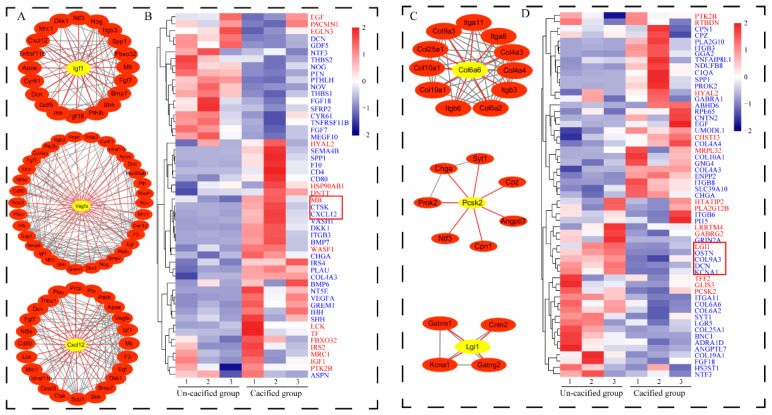
Map of protein–protein interaction (PPI) networks and a gene expression cluster analysis between the secretory proteins and differential expressed genes. (**A**) PPI network between the sternum secretory protein and the pectoral muscle differential gene. (**B**) Gene expression cluster of all genes selected from PPI networks (Figure 7A). Different groups of genes are distinguished by different colors, in which red represents DEGs from muscles and blue represents secretory proteins from bones. (**C**) PPI network between the proteins secreted from the pectoral muscle and the differential expressed genes in the calcified and un-calcified sternum. (**D**) Gene expression cluster analysis of all genes selected from PPI networks (Figure 7C). Different groups of genes are distinguished by different colors, in which red represents secretory proteins from muscles and blue represents DEGs from bones.

**Figure 8 genes-12-00463-f008:**
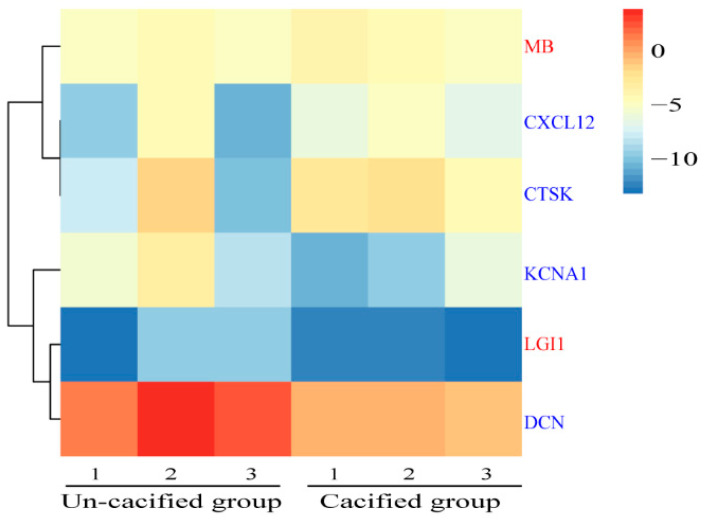
The expression patterns of some differential expression genes (DEGs) in muscles and bones detected by RT-PCR. The gene expression values are transformed to a log2 scale. Different groups of genes are distinguished by different colors, in which red represents the labels of DEGs from muscles and blue represents the labels of secretory proteins from bones.

## Data Availability

The datasets generated for this study can be found in the NCBI SRA (accession number PRJNA605777).

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
