# Peer review of "Evidence Supporting the Regulatory Relationships through a Paracrine Pathway between the Sternum and Pectoral Muscles in Ducks"

_genes, 2021, doi:10.3390/genes12040463_

Round 1
Reviewer 1 Report
Line 221: Change the period that separates the sentence to a comma. Change " ... and muscle. We performed" by "... and muscle, we performed".
Figure 8. The colors red and blue refer to the labels of the names of the genes, or the colors of the cells.
Line 280: Change "DGEs" by "DEGs"
Reviewer 2 Report
By RNA-seq, the authors demonstrated that the pectoral muscle development was affected by the sternal 25 ossification states in ducks. The VEGFA, CXCL12, SPP1, NOG, and BMP7 were possibly the key 26 genes to participate in the ossification of duck sternum. They firstly listed evidence supporting the 27 regulatory relationships through a paracrine pathway between the sternum and pectoral muscles 28 in ducks, which provided scientific data for the study of the synergistic development of bone and 29 skeletal muscle. The study is potential novel and my concerns have been addressed.
Author Response
Please see the attachment.

This manuscript is a resubmission of an earlier submission. The following is a list of the peer review reports and author responses from that submission.
Round 1
Reviewer 1 Report
The authors of this manuscript studied the relationships between sternum and pectoral muscle in ducks, which is a little studied subject. Although there is little information on the subject, the introduction presents facts reported for mammals instead of birds. I recommend an extension as much as possible of bird-focused references.
In Results section, the figure 4A is a bit confusing for me. I expected a correlation of 1.0 between equal samples, but the highest correlation occurs between different samples. Are sample names well accommodated? The diagonal with values of 1.0 must be between samples that are equal.
There should be a little more clarity regarding whether the reported DEGs are in the calcified or un-calcified group. Similarly, Figure 4B should indicate the meaning of the colors of the bars.
Figure 5A is a bit confusing with respect to the bars. I understand that black line bars are the percentage of genes associated with GO, and that red line bars are what indicate the p-value. I advise to separate the data and in a graph represent the percentages and in another the p-value.
Figure 7B and 7D show that there are genes whose expression pattern changes within the same groups. For example, the PTK2B, IGF1, MRC1 genes, of Figure 7B within the un-calcified group in sample 1 are being expressed while in sample 2 and 3 they are repressed. Which leads to infer that there is no consistent expression pattern within the samples of the same group, how can you explain this?
The evidence on the regulatory relationships between the sternum and the pectoral muscle must be expanded, since the results only demonstrate a consistent pattern of expression between LGI1 and KCNA1 genes obtained from the RNA-seq analysis. Why was a RT-PCR of these genes not done to punctually evaluate their expression pattern?
Line 175. It says "molecular function" and must be "cellular component"
Line 178. It says that there are 13 enriched KEGG routes, but in graph 5B there are only 12.
Line 195. Were the 195 genes that were used in GeneCards up-regulated?
Reviewer 2 Report
The bone and muscle are two independent and largest secretion organs (endocrine function), which are synergistic in vertebrate’s development. Investigating the coordination development process between bone and skeletal muscle would be a new perspective way for the therapy of human sarcopenia and osteopenia. The authors suggest that the pectoral muscle development was affected by the sternal ossification states in ducks. The VEGFA, CXCL12, SPP1, NOG and BMP7 were possibly the key genes to participate in the ossification of duck sternum. We firstly listed evidence supporting the regulatory relationships through a paracrine pathway between the sternum and pectoral muscles in ducks, which provided scientific data for the study of the synergistic development of bone and skeletal muscle.
Major concerns
Why did the authors want to study the relationship between bone and skeletal muscle? Please provide more evidence in Introduction section. Why did the authors use duck model? The authors should deeply analyze the novel differentially expressed genes, and RT-PCR verification should be done. The function roles of VEGFA, CXCL12, SPP1, NOG and BMP7 should be verified in disease model, if possible. Ethical statement is needed, because of animal model.Author Response
Please see the attachment.
